# Syne Tune: A Library for Large-Scale Hyperparameter Tuning and Reproducible Research

David Salinas[1]  Matthias Seeger[1]  Aaron Klein[1]  Valerio Perrone[1]  Martin Wistuba[1]
Cedric Archambeau[1]

[1]Amazon AWS

**Abstract**  We present Syne Tune, a library for large-scale distributed hyperparameter optimization (HPO).[1] Syne Tune's modular architecture allows users to easily switch between different execution backends to facilitate experimentation and makes it easy to contribute new optimization algorithms. To foster reproducible benchmarking, Syne Tune provides an efficient simulator backend and a benchmarking suite, which are essential for large-scale evaluations of distributed asynchronous HPO algorithms on tabulated and surrogate benchmarks. We showcase these functionalities with a range of state-of-the-art gradient-free optimizers, including multi-fidelity and transfer learning approaches on popular benchmarks from the literature. Additionally, we demonstrate the benefits of Syne Tune for constrained and multi-objective HPO applications through two use cases: the former considers hyperparameters that induce fair solutions and the latter automatically selects machine types along with the conventional hyperparameters.

## 1 Introduction

Major advances in algorithms, systems, and hardware led to deep learning models with billions of parameters being trained by gradient-based stochastic optimization. However, these algorithms come with many hyperparameters that are crucial for good performance. Hyperparameters come in many flavors, such as the learning rate and its schedule, the type and amount of regularization or the number and width of neural network layers. Tuning them is difficult and time-consuming, even for experts, criteria like latency or cost often play a role in deciding for a winning hyperparameter configuration. If domain experts and industrial practitioners are to benefit from latest deep learning technology, it is essential to automate the *tuning* of these hyperparameters along with speeding up the *training* of neural network weights.

Today, a number of *hyperparameter optimization* (HPO) systems are available in order to fill this need, both commercially (Golovin et al., 2017; Perrone et al., 2021b) and open source (Liaw et al., 2018; Akiba et al., 2019; Lindauer et al., 2021). Taken together, they reflect the wide diversity of this research area, offering different optimization strategies, evaluating hyperparameter configurations sequentially or in parallel, and enabling tuning single hyperparameters to full-blown neural architecture search (NAS). Some are *single-fidelity* optimizers that only rely on evaluations obtained after full training runs, while others allow to make a more efficient use of compute by early stopping the training process; we will refer to these as *multi-fidelity* optimizers as they also make use of "low fidelity" evaluations along the training process. However, few systems support advanced settings such as constrained, multi-objective or transfer learning-based HPO.

This diversity can be hard to navigate for non-expert users, but also for HPO researchers. Most available systems are not agnostic along axes such as HPO algorithm or execution backend (which runs the training jobs), but only cover parts of the whole. Moreover, tooling for empirical comparisons and benchmarking is often sidelined. In contrast, when new ideas for *training* deep

---

[1]https://github.com/awslabs/syne-tune

neural networks emerge today, they can be compared against others in the same system with the same backend, such as *PyTorch* (Paszke et al., 2019) or *TensorFlow* (Abadi et al., 2015). The field of *tuning* such large models could equally benefit from systems that reduce these confounding factors.

In this paper we present *Syne Tune*, a novel open source library for asynchronous distributed HPO. The design of Syne Tune addresses issues of implementation and system bias, and it comes with tooling to simplify and speed up empirical comparisons of HPO methodology. The main features of Syne Tune are the following:

- Wide coverage of baselines: Syne Tune provides implementations across the spectrum of HPO algorithms, such as random search, Bayesian optimization and evolutionary search, as well as a range of asynchronous and synchronous multi-fidelity optimizers, thus removing the implementation bias in comparisons. See Table 3 in the Appendix for a list of currently supported algorithms.

- Backend agnostic: Syne Tune allows to easily switch the execution backend. Coming with a generic API, new backends can be easily integrated. As of now, Syne Tune provides three backends to change seamlessly the execution locally, on the cloud or as simulation. In particular, the *simulator backend* allows to run realistic experiments on a single CPU, where the actual runtime is determined by the decision-making only.

- Advanced HPO methodologies: beyond the basic global optimization formulation of single- and multi-fidelity HPO, Syne Tune supports a range of advanced setups, such as hyperparameter transfer learning, constrained HPO or multi-objective optimization.

- Benchmarking: Syne Tune provides an abstraction for benchmarks, along with a sizeable suite of implementations thereof. A special emphasis is given to supporting fast, affordable experimentation with *tabulated* or *surrogate* benchmarks and a simulation backend.

## 2 Related Work

Recently, a large variety of HPO frameworks have emerged. Due to space constraints, we only discuss frameworks for hyperparameter optimization and omit other frameworks for general AutoML, such as Auto-Sklearn (Feurer et al., 2020) or AutoGluon (Erickson et al., 2020). Arguably, most similar to Syne Tune is Ray Tune (Liaw et al., 2018), which uses Ray (Moritz et al., 2017) as the backend to distribute the HPO process. While Ray Tune supports a range of HPO algorithms, it does not provide support for multi-objective optimization, constrained optimization, transfer learning or benchmarking. Optuna (Akiba et al., 2019) accommodates TPE (Bergstra et al., 2011), CMA-ES (Hansen, 2006) and even multi-objective optimization, but lacks recent multi-fidelity approaches, such as ASHA (Li et al., 2019) or PBT (Jaderberg et al., 2017). SMAC3 (Lindauer et al., 2021) offers multi-fidelity algorithms, such as BOHB (Falkner et al., 2018) or Hyperband (Li et al., 2018), alongside Bayesian optimization (BO), but does not support multi-objective or constrained optimization. It also lacks system support for asynchronous parallel algorithms, such as checkpointing of neural networks or asynchronous scheduling. Dragonfly (Kandasamy et al., 2020) provides BO based strategies for distributed HPO, but does not support multi-objective and transfer learning scenarios. Hyper-Tune (Li et al., 2022) contains a range of asynchronous multi-fidelity algorithms based on successive halving, but does not support distributed tuning across multiple machines, nor does it provide integrated benchmarks for reproducible large-scale experiments. Loosely related are also BO packages, such as BOTorch (Balandat et al., 2020). However, their main purpose is to provide a platform to foster research on Monte Carlo BO rather than distributed HPO.

In an orthogonal line of work, several packages support efficient evaluation of HPO algorithms by providing a standardized access to benchmarks. HPOBench (Eggensperger et al., 2021) serves code for a range of multi-fidelity benchmarks and comes with a thorough evaluation of state-of-the-art approaches, which however does not include transfer learning methods. HPO-B (Arango

```
1   # hyperparameter search space to consider
2   config_space = {
3       'epochs': 100,
4       'num_layers': randint(1, 20),
5       'learning_rate': loguniform(1e-6, 1e-4)
6   }
7   tuner = Tuner(
8       trial_backend=LocalBackend(entry_point='train_network.py'),
9       scheduler=BayesianOptimization(config_space, metric='val_loss'),
10      stop_criterion=StoppingCriterion(max_wallclock_time=30),
11      n_workers=4,  # how many trials are evaluated in parallel
12  )
13  tuner.run()
```

Listing 1: How to tune a training script with Bayesian optimization in Syne Tune.

et al., 2021) contains a set of surrogate benchmarks (Eggensperger et al., 2015) based on datasets from OpenML (Vanschoren et al., 2014). They also compare several state-of-the-art hyperparameter transfer learning algorithms in the single-fidelity setting. Mehta et al. (2022) introduced NASLib which implements several search spaces and optimization strategies for neural architecture search. To the exception of Li et al. (2022) which provides an experimental script to simulate tabulated benchmarks from NAS201, none of these packages allow to efficiently simulate asynchronous parallel methods on multiple tabular and surrogate benchmarks.

## 3 Library Overview

Let $x$ be a configuration of hyperparameters in the space $\mathcal{X}$. Our goal is to find a global minimum of the target function $f(x)$, which could be non-differentiable. We will further consider a *multi-fidelity* setting, assuming that an evaluation, which can be noisy, requires the consumption of the maximum resource level $r_{\max}$. The optimization problem of interest is defined as follows:

$$\min_{x \in \mathcal{X}} f(x, r_{\max}).$$

Observations at $r < r_{\max}$ are cheaper, yet may be only loosely correlated with $f(x, r_{\max})$. To fix ideas, $f(x, r)$ could for instance be the validation error of a neural network trained for $r$ epochs with $x$ encoding the learning rate, batch size and number of units of a layer.

Next, we introduce the core modules of Syne Tune: the tuner, the backends, the schedulers and the benchmarking components.

### 3.1 Tuner

The overall search for the best configuration is orchestrated by the *tuner*. It interacts with a *backend*, which launches new *trials* (i.e., evaluations of configurations) in parallel and retrieves evaluation results. Decisions on which trials to launch or interrupt are delegated to a *scheduler*, which is synonymous with "HPO algorithm" in Syne Tune. A code snippet for launching an HPO experiment based on BO in Syne Tune is given in Listing 1. Algorithm 1 in the Appendix contains pseudo-code for the tuner loop.

When a worker is free, the tuner queries the scheduler for a trial, passing it to the backend for execution. Usually, the scheduler searches for a new, most promising configuration $x$, but for some schedulers, a paused trial may be resumed instead. Whenever a trial reports evaluations $f(x, r)$, they are sent to the scheduler, who may use this data to improve future decisions. The

scheduler returns a decision on the reporting trial (stop, pause or continue), which is executed by the backend.

During a tuning experiments, results obtained over time and the tuner state are periodically stored. The former allow for live plotting of relevant metrics (e.g., best performance attained so far). By loading the tuner state, tuning can be resumed later on. The ability to recover from premature termination allows us to work with cheaper, preemptible compute instances. A sound comparison of several different combinations of schedulers requires averaging results over a substantial number of runs with different random seeds (Agarwal et al., 2021), since the outcomes of single HPO experiments can be highly variable. Syne Tune allows for this effort to be parallelized by running tuning remotely.

## 3.2 Backends

The backend module is responsible for starting, stopping, pausing and resuming trials and accessing results and trial statuses. Syne Tune provides a general interface for a backend and three implementations: one to evaluate trials on a local machine, one to evaluate trials on the cloud, and one to simulate tuning with tabulated benchmarks to reduce run time. Switching between different backends can be done by just passing a different `trial_backend` parameter of the tuner as shown in the code examples given in Appendix D. The backend API has been kept lean on purpose, and adding new backends requires little effort. Note that pause-and-resume scheduling requires checkpointing of models, which is supported by all backends in Syne Tune.

**Local backend**. This backend evaluates trials concurrently on a single machine by using subprocesses. We support rotating multiple GPUs on the machine, assigning the next trial to the least busy GPU, e.g. the GPU with the smallest amount of trials currently running. Trial checkpoints and logs are stored to local files.

**Cloud backend**. Running on a single machine limits the number of trials which can run concurrently. Moreover, neural network training may require many GPUs, even distributed across several nodes (Brown et al., 2020). For those use-cases, we provide an Amazon SageMaker backend that schedules one training job per trial. Amazon SageMaker provides several benefits for conducting reproducible HPO research (Liberty et al., 2020). Users have access to pre-build containers of ML frameworks (e.g., Pytorch, Tensorflow, Scikit-learn, HuggingFace), training on cheaper preemptible machines and distributed training work out-of-the-box. We demonstrate the versatility of this backend in Section 4.3.

**Simulation backend**. A growing number of tabulated benchmarks are available for HPO and NAS research (Ying et al., 2019; Dong and Yang, 2020; Klein et al., 2019; Klein and Hutter, 2019; Siems et al., 2020; Klyuchnikov et al., 2020; Arango et al., 2021). The simulation backend allows to run realistic experiments with such benchmarks on a single CPU instance, paying real time for the decision-making only. To this end, we use a time keeper to manage simulated time and a priority queue of time-stamped events (e.g., reporting metric values for running trials), which work together to ensure that interactions between trials and scheduler happen in the right ordering, whatever the experimental setup may be. The simulator correctly handles any number of workers, and delay due to model-based decision-making is taken into account.

## 3.3 Schedulers

In Syne Tune, HPO algorithms are called *schedulers*. They interact with the tuner by suggesting configurations for new trials, but may also decide to stop running trials, or resume paused ones. All currently supported schedulers are listed in Table 3, and in the sequel, we restrict our focus on these, noting that there is a lot more related work in any of the directions we touch upon here; see for instance the comprehensive review by Feurer and Hutter (2019).

All our schedulers suggest single new configurations in the presence of pending evaluations (i.e., running trials which have not reported results yet). Decision-making is synchronized in some schedulers, such as Hyperband (Li et al., 2018) or synchronous BOHB (Falkner et al., 2018), where trials have to wait in order to get resumed (or terminated) until a number of others reached their resource level. Most of our schedulers make decisions asynchronously (i.e., on the fly), which is more efficient in general (Li et al., 2018; Klein et al., 2020).

In multi-fidelity HPO, we receive evaluations $f(x, r)$ at resource levels $r \in \{r_{\min}, \ldots, r_{\max}\}$ (e.g., epochs) and may act on these by stopping or resuming trials. ASHA (Li et al., 2019), a prominent asynchronous multi-fidelity algorithm, comes in two variants (Klein et al., 2020): *stopping* stops underperforming trials early, while *promotion* pauses trials and resumes the most promising ones later on, but requires model checkpointing.

Finally, configurations for new trials can be proposed by random sampling (Bergstra et al., 2011), Bayesian optimization (BO) (Snoek et al., 2012) or evolutionary techniques (Real et al., 2019). BO schedulers fit a probabilistic surrogate model to the evaluation data and propose the maximizer of an acquisition function averaged over the posterior distribution of this model. This often leads to more sample efficiency, but expensive decisions can prolong experiment wall-clock time. The combination of BO with multi-fidelity is non-trivial, because $f(x, r)$ needs to be modelled along $x$ and $r$. Different options are given in (Swersky et al., 2014; Klein et al., 2020; Li et al., 2022).

**Transfer learning-based schedulers**. Hyperparameter transfer learning aims to use evaluation data from past HPO tasks in order to warmstart the current HPO task (Bardenet et al., 2013; Wistuba et al., 2015a; Perrone et al., 2019b; Salinas et al., 2021b). Syne Tune supports transfer learning-based HPO via an abstraction which maps a scheduler and transfer learning data to a warmstarted instance of the former. In our experiments, we consider ASHA-BB, which implements the bounding-box method of Perrone et al. (2019b), ASHA-CTS, which combines the quantile-based approach of Salinas et al. (2021b) with ASHA, and ZS which greedily selects hyperparameter configurations that complement previously considered ones based on performances on other tasks (Wistuba et al., 2015b). RUSH (Zappella et al., 2021) is another Syne Tune scheduler which warmstarts ASHA with best configurations of previous tasks. Additionally, it prunes all configurations which don't outperform the initial configurations.

**Constrained schedulers**. Constrained HPO amounts to the problem $\min_{x \in \mathcal{X}} f(x)$, subject to $c(x) \leq 0$ (Gardner et al., 2014), where the constraint function $c(x)$ needs to be learned by sampling just like $f(x)$. Syne Tune provides the the approach from Gardner et al. (2014), called CBO in Table 3, where $f(x)$ and $c(x)$ have independent Gaussian process surrogate models, coming together in an expected constrained improvement acquisition function.

**Multi-objective schedulers**. The goal of multi-objective HPO (Emmerich and Deutz, 2018) is to find good trade-offs between several objectives $f_1, \ldots, f_k$. A Pareto-optimal point $x \in \mathcal{X}$ cannot be improved upon along any one objective without increasing along another one. Formally, for any $x' \in \mathcal{X}$, $k_1$, if $f_{k_1}(x') < f_{k_1}(x)$, there is some $k_2$ such that $f_{k_2}(x') > f_{k_2}(x)$. We would like to sample configurations on the Pareto frontier (i.e., the set of Pareto-optimal points). MOASHA is a multi-objective extension of ASHA, where trials paused at each rung level are ranked by non-dominated sorting (Salinas et al., 2021a; Guerrero-Viu et al., 2021; Schmucker et al., 2021).

### 3.4 Benchmarking

Similar to HPOBench or HPO-B, we provide benchmarking functionalities to simplify and standardize comparisons of HPO algorithms. However, we surpass previous efforts in terms of speed of experimentation, both, when it comes to running experiments in parallel and to simulating them for tabulated benchmarks.

Table 1: Performance comparison when loading a task and reading all fidelities of a random configuration. These tasks can have a significant impact on the runtime of simulated benchmarks.

| | load time (s) | | | read time (s) | | |
|---|---|---|---|---|---|---|
| | FCNet | NAS201 | LCBench | FCNet | NAS201 | LCBench |
| Baseline | 7.5892 | 31.9592 | 8.8561 | 0.2898 | 0.2274 | 5e-6 |
| Syne Tune | 0.7443 | 0.2356 | 0.2528 | 0.0005 | 0.0004 | 5e-6 |

Syne Tune contains a repository, where tabulated benchmarks are stored and served in a single general format optimized for fast look-up and reading time. Table 1 shows the average runtime needed to load a tabulated benchmark (averaged over 3 seeds), and the average runtime needed to access all fidelities of a random configuration (averaged over 1000 reads), compared to publicly available implementations (FCNet and NAS201 from Eggensperger et al. (2021), LCBench from Zimmer et al. (2021)). Speeding up the access to look-up table values can result in simulated experiments to complete much faster. For instance, to simulate one scheduler on 30 seeds on the 4 tasks of FCNet, roughly 600K evaluations are needed which corresponds to 5 min of total reading time on a single machine with Syne Tune versus 2 days with the best public available implementation of FCNet.

Ultimately, faster experimentation leads to more robust results, due to more repetitions with different random seeds, and more alternatives can be considered in the time available to the scientist.

## 4 Experiments

In this section, we first conduct a large scale comparison of asynchronous algorithms, including multi-fidelity and hyperparameter transfer learning, and then study two use-cases: constrained BO for improving fairness in a ML pipeline, and multi-objective optimization to jointly tune hardware and hyperparameter configurations.

### 4.1 Comparing asynchronous HPO algorithms

**Baselines**. We consider single-fidelity algorithms RS, GP and REA, and multi-fidelity algorithms ASHA, asynchronous BOHB, BORE, MSR and MOB, detailed in Section 3.3 and Table 3. We also evaluate hyperparameter transfer learning methods ASHA-BB, ASHA-CTS, RUSH and ZS (see Section 3.3). When running on a given task, they are provided all evaluations from the others tasks of the same benchmark as transfer data.

**Benchmarks**. We use 12 problems coming from the tabulated benchmarks FCNet (Klein and Hutter, 2019), NAS201 (Dong and Yang, 2020) and LCBench (Zimmer et al., 2021). Details on these benchmarks and their configuration spaces are given in Appendix B.

**Experiment details**. Unless otherwise stated, tuning experiments use the simulation backend on a AWS c5.4xlarge machine with 4 parallel workers, and are stopped when a wall-clock time budget is exhausted. All runs are repeated 30 times with different random seeds, and means and standard deviations are reported. Scripts to reproduce all our experiments together with instructions are available in this repository [2].

**Simulating parallel tuning**. In our first experiment, we compare the performance of ASHA with an increasing number of workers to execute jobs asynchronously in parallel (Figure 1 left). The major speed-up in wall-clock time obtained by running multiple workers asynchronously is a key aspect that is leveraged in industrial applications (Li et al., 2019; Klein et al., 2020).

---

[2]https://github.com/awslabs/syne-tune/tree/main/benchmarking/nursery/benchmark_automl

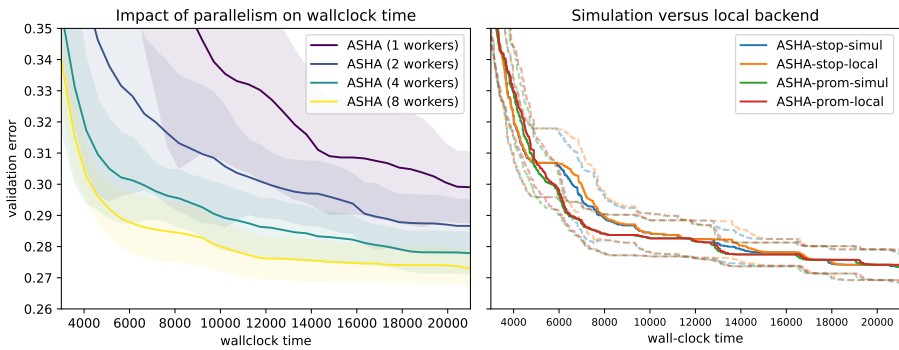

Figure 1: **Left**: Effect of number of workers on performance when running ASHA on NAS201 (CIFAR-100). The time to reach 30% validation accuracy with one worker gets reduced by respectively 1.8, 3 and 4 when using 2, 4, and 8 workers. **Right**: Comparison of simulator backend with local backend on the same task (10 random repetitions). For the local backend, workers are put to sleep for the tabulated training time. For the simulator backend, real wall-clock times were 17.7($\pm$4.6) seconds for ASHA stopping, 17.5($\pm$3.1) seconds for ASHA promotion.

In order to test whether the simulation backend provides realistic results, we compared against the naive approach of putting each worker to sleep for the tabulated training time. As shown in Figure 1 right, quantitative differences between the two alternatives are negligible. However, while the naive approach takes the full 6.25 hours, the simulated results are available in about 17 seconds.

**Comparison of asynchronous algorithms**. In Figure 2, we show the performance of all optimizers on two tasks. Results are similar for other tasks, as shown in Appendix C. Table 2 shows the ranks of different algorithms, averaged over time, seeds and benchmarks.

REA and TPE perform similar to RS, despite them exploiting past information. GP does better on FCNet (slice). Multi-fidelity algorithms are generally superior on these tasks, as is known in the field (Li et al., 2018, 2019; Klein et al., 2020; Eggensperger et al., 2021). Among them, MSR is the only one not using successive halving, and it performs worst. Using past data via Bayesian optimization, MOB has an edge on ASHA and asynchronous BOHB on NAS201, yet the three perform similarly on FCNet. While the use of advanced asynchronous scheduling via successive halving reduces the benefits of BO decision-making , compared to GP versus RS, it should be noted that our experimental setup is rather favourable for ASHA. More complex execution backends, spanning several machines, can come with substantial delays for starting or stopping jobs. This increases the relative cost of starting a job, even if it is stopped after one epoch, and the strategy of starting randomly drawn configurations becomes less attractive.

Finally, the transfer learning approaches are provided with evaluation data from related tasks, not available to the others. This allows them to significantly improve the performance. Depending on the application, transfer data may be available or not. In practice, it may require an infrastructure for logging, aggregation, and cleaning. Syne Tune allows to mix in such data in various ways, so that any promising HPO algorithm in the standard setting can easily be enhanced to make use of valuable data from past experiments.

### 4.2 Constrained hyperparameter optimization use case

ML applications often require a solution satisfying pre-defined constraints, such as maximum memory usage, training cost, inference latency or even fairness (Perrone et al., 2019a, 2021a; Lee et al., 2020; Guinet et al., 2020). For example, a classifier that is only tuned to maximize prediction accuracy may unfairly predict a high credit risk for some subgroups of the population applying for

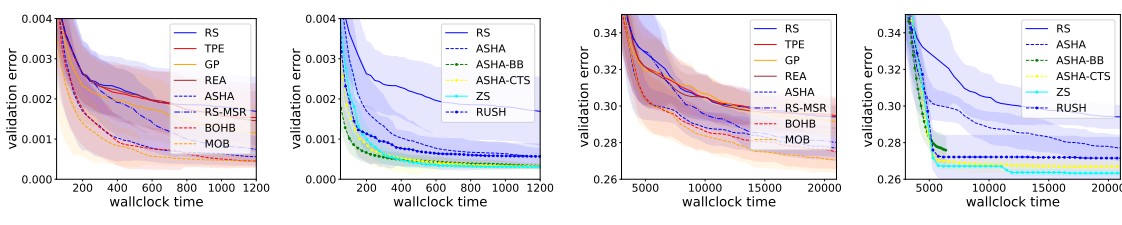

(a) FCNet: Slice Localization.

(b) NAS201: CIFAR-100.

Figure 2: Best validation error found over time on (a) FCNet (slice) and (b) NAS201 (CIFAR-100). Results averaged over 30 random repetitions. For each benchmark **Left**: single- and multi-fidelity methods; **Right**: transfer learning methods.

Table 2: Average normalized rank (lower is better) of algorithms across time and benchmarks. Best results per category are indicated in bold.

|         | single-fidelity | | | | multi-fidelity | | | | transfer-learning | | | |
|---------|------|------|------|------|------|------|------|------|------|--------|------|----------|
|         | RS   | TPE  | REA  | GP   | MSR  | ASHA | BOHB | MOB  | RUSH | ASHA-BB | ZS   | ASHA-CTS |
| FCNet   | 0.78 | 0.72 | 0.66 | **0.60** | 0.61 | 0.47 | 0.47 | **0.36** | 0.58 | 0.29 | 0.28 | **0.22** |
| NAS201  | 0.76 | **0.74** | 0.73 | 0.75 | 0.67 | 0.49 | 0.49 | **0.48** | 0.31 | 0.27 | **0.13** | 0.20 |
| LCBench | 0.68 | 0.67 | 0.68 | **0.57** | 0.50 | 0.46 | **0.43** | 0.50 | **0.23** | 0.43 | 0.53 | 0.31 |
| Average | 0.74 | 0.71 | 0.69 | **0.64** | 0.59 | 0.47 | **0.46** | **0.45** | 0.38 | 0.33 | 0.31 | **0.24** |

a loan. In this section, we show how Syne Tune's constrained HPO can be used to optimize for accuracy while enforcing additional fairness constraints.

We consider the German Credit Data from the UCI Machine Learning Repository (Dua and Graff, 2017), a binary classification problem where the goal is to classify people described by a set of attributes as good or bad credit risks. One of these attributes is a binary variable indicating gender, a sensitive feature. We measure unfairness as the deviation from statistical parity (DSP) between individuals of different genders, namely $|P(\hat{Y} = 1|S = 0) - P(\hat{Y} = 1|S = 1)|$, where $Y$ is the true label, $S$ the sensitive attribute, and $\hat{Y}$ the predicted label. We tune a Random Forest model to optimize validation accuracy with a random 70%/30% split into train/validation, under the constraint of DSP being lower than 0.01. Syne Tune implements the constrained EI (cEI), a popular acquisition function for constrained Bayesian optimization (cBO) (Gardner et al., 2014).

Figure 3 (left) provides a comparison of standard BO and cBO, each run for 15 iterations. For every trial, we report both the DSP and validation accuracy level. The results show that standard BO can get stuck in high-performing yet unfair regions, failing to return a well-performing, feasible

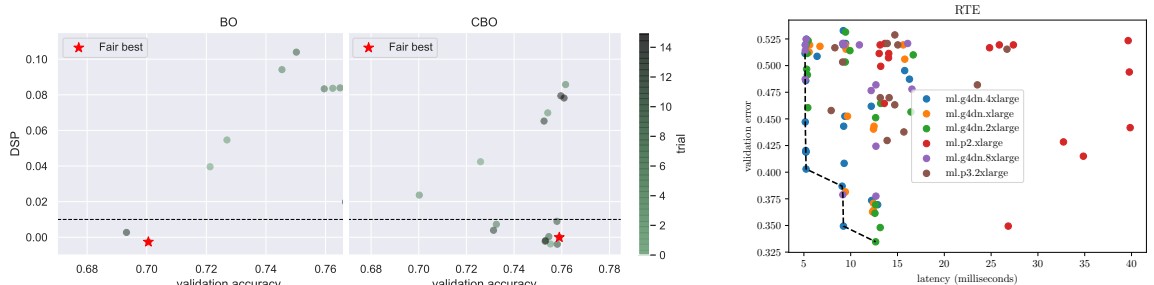

Figure 3: **Left**: Comparison of BO and cBO for tuning a random forest on German. The horizontal line is the fairness constraint (DSP ≤ 0.01). **Right**: Latency and test error of configurations sampled by MOASHA. The dashed black line represents the Pareto front, points on which constitute optimal trade-offs.

solution. In contrast, Syne Tune's constrained BO focuses the exploration over the fair area of the hyperparameter space and finds a more accurate fair solution.

### 4.3 Multi-objective hyperparameter optimization use case

Our final example shows what can be done by combining Syne Tune's multi-objective optimizers with its cloud backend. In industrial practice, predictive performance is rarely the only metric of concern, prediction latency, training time and cost are often at least equally important. Moreover, every user may trade them off differently. Using a cloud service for training ML models, many different machine types, spanning a large range of specs and costs, can be accessed by just changing an argument. We show how this complexity can efficiently be navigated by including the machine type among the parameters to be tuned with multi-objective HPO. While the machine type does not influence predictive performance, it strongly impacts latency, training time and cost. We use MOASHA (Section 3.3, Table 3) in order to sample the Pareto set for a number of such metrics. Training jobs are executed with the Amazon SageMaker backend.

We demonstrate the impact of jointly optimizing instance type together with model hyperparameters when fine-tuning a large pre-trained language model from HuggingFace (Wolf et al., 2019) on the RTE dataset of the GLUE benchmark suite (Wang et al., 2019). As hyperparameters we expose the learning rate, batch size, the warm-up ratio of a triangular learning rate schedule as well as the choice of the pre-trained model. We use the Amazon SageMaker backend in order to distribute HPO across a heterogenous set of instances. Figure 3 (right) shows the latency and test error of all configurations sampled by MOASHA after a budget of 1800 seconds with 4 workers. Selecting the right instance type correlates with other hyperparameters. For example, while `g4dn.4xlarge` instances tend to have better latency than `p3.2xlarge` instances, for some configurations other instance types still seem to work even better.

## 5 Discussion

Syne Tune is an open source library for distributed HPO with an emphasis on enabling reproducible machine learning research. It simplifies, standardizes and speeds up the evaluation of a wide variety of HPO algorithms, which are implemented on top of common modules and aim to remove implementation bias to conduct fair comparisons. By supporting different backends, Syne Tune lets researchers and engineers effortlessly move from simulation and small-scale experimentation to large-scale distributed tuning on the cloud. Its simulator backend, curated set of benchmarks, and problem repository allow researchers to conduct robust empirical evaluations with many random repetitions, rapidly and at low costs, in particular by taking advantage of the growing availability of tabulated benchmarks. Beyond the basic single- and multi-fidelity HPO, Syne Tune supports advanced scenarios such as constrained HPO, multi-objective optimization, and hyperparameter transfer learning with the goal to offer a better coverage of academic and industrial use cases.

## 6 Limitations and Broader Impact Statement

Syne Tune has the potential to make automated tuning research more efficient, reliable, and trustworthy. We also hope it will encourage domain experts to adopt HPO methodology more systematically. By making simulation on tabulated benchmarks a first class citizen, it allows researchers without massive computation budgets to participate. However, Syne Tune may foster an over-reliance on tabulated benchmarks, which are often smaller and less variable than real benchmarks that need computation. In general, efforts to collect and publish new benchmarks and diverse data sets are of critical importance to assess advances in the field, and should complement endeavours targeted at rendering machine learning research more reproducible.

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

**Algorithm 1:** Pseudo-code of the tuner loop.

```
1  while not stop_criterion() do
2  │   new_results = backend.fetch_results() ;
3  │   for result in new_results do
4  │   │   decision = scheduler.on_trial_result(result) ;
5  │   │   if decision == 'stop' then
6  │   │   │   backend.stop_trial(result.trial_id) ;
7  │   │   end
8  │   end
9  │   if backend.num_free_workers() > 0 then
10 │   │   while backend.num_free_workers() > 0 do
11 │   │   │   config = scheduler.suggest() ;
12 │   │   │   backend.start_trial(config) ;
13 │   │   end
14 │   else
15 │   │   time_keeper.advance(sleep_time) ;
16 │   end
17 end
```

## A  Additional library details

### A.1  Schedulers

We list the schedulers currently supported in Syne Tune, see Section 3.3 for details. TPE search uses kernel density estimators in order to approximate its acquisition function directly (Falkner et al., 2018). PBT differs from the others, in that a trial can change its configuration, and can be resumed from the checkpoint of another trial.

Table 3: Schedulers supported in Syne Tune.

| name | reference | (a)sync | multi-fidelity | search |
|---|---|---|---|---|
| RS | Bergstra et al. (2011) | async | no | random |
| GP | Snoek et al. (2012) | async | no | BO |
| TPE | Bergstra et al. (2011) | async | no | density-ratio estimation |
| REA | Real et al. (2019) | async | no | evolution |
| BORE | Tiao et al. (2021) | async | no | density-ratio estimation |
| ASHA | Li et al. (2019) | async | stopping/promotion | random |
| MOB | Klein et al. (2020) | async | stopping/promotion | BO |
| BOHB | Falkner et al. (2018) | async | stopping/promotion | TPE |
| MSR | Golovin et al. (2017) | async | stopping | random |
| SyncHyperband | Li et al. (2018) | sync | promotion | random |
| SyncBOHB | Falkner et al. (2018) | sync | promotion | TPE |
| PBT | Jaderberg et al. (2017) | sync | promotion | evolution |
| RUSH | Zappella et al. (2021) | async | stopping/promotion | random |
| ASHA-BB | Perrone et al. (2019b) | async | stopping/promotion | random |
| ASHA-CTS | Salinas et al. (2021b) | async | stopping/promotion | parametric surrogate |
| ZS | Wistuba et al. (2015b) | async | no | greedy-selection |
| MOASHA | Schmucker et al. (2021) | async | stopping | random |
| CBO | Gardner et al. (2014) | async | no | BO (CEI) |

### A.2  Tuner

In Algorithm 1, we provide a stylized version of what happens in the tuner loop. Here, `time_keeper.advance(sleep_time)` sleeps for the alloted time in a standard backend, but in this simulation backend, the time keeper is simply just advanced (see Section 3.2).

## B  Experiment details

Statistics for tabulated benchmarks are given in Table 4 and their configuration spaces are given in Table 5. The domains "finite-range" and "finite-range log-space" correspond to `finrange` and

Table 4: Tabulated benchmark statistics

| Benchmark | #Evaluations | #Hyperparameters | #Tasks | #Fidelities |
|-----------|--------------|------------------|--------|-------------|
| FCNet     | 62208        | 9                | 4      | 100         |
| NAS201    | 15625        | 6                | 3      | 200         |
| LCBench   | 2000         | 7                | 5      | 52          |

Table 5: Configuration spaces for all tabulated benchmarks.

| Benchmark | Hyperparameter | Configuration space | Domain |
|-----------|----------------|---------------------|--------|
| FCNet | activation_1 | ["tanh", "relu"] | categorical |
|       | activation_2 | ["tanh", "relu"] | categorical |
|       | batch_size   | [8, 16, 32, 64] | finite-range log-space |
|       | dropout_1    | [0.0, 0.3, 0.6] | finite-range |
|       | dropout_2    | [0.0, 0.3, 0.6] | finite-range |
|       | init_lr      | [0.0005, 0.001, 0.005, 0.01, 0.05, 0.1] | categorical |
|       | lr_schedule  | ["cosine", "const"] | categorical |
|       | n_units_1    | [16, 32, 64, 128, 256, 512] | finite-range log-space |
|       | n_units_2    | [16, 32, 64, 128, 256, 512] | finite-range log-space |
| NAS201 | x0 | ["avg_pool_3x3", "nor_conv_3x3", "skip_connect", "nor_conv_1x1", "none"] | categorical |
|        | x1 | ["avg_pool_3x3", "nor_conv_3x3", "skip_connect", "nor_conv_1x1", "none"] | categorical |
|        | x2 | ["avg_pool_3x3", "nor_conv_3x3", "skip_connect", "nor_conv_1x1", "none"] | categorical |
|        | x3 | ["avg_pool_3x3", "nor_conv_3x3", "skip_connect", "nor_conv_1x1", "none"] | categorical |
|        | x5 | ["avg_pool_3x3", "nor_conv_3x3", "skip_connect", "nor_conv_1x1", "none"] | categorical |
| LCBench | num_layers    | [1, 5]       | uniform |
|         | max_units     | [64, 512]    | log-uniform |
|         | batch_size    | [16, 512]    | log-uniform |
|         | learning_rate | [1e-4, 1e-1] | log-uniform |
|         | weight_decay  | [1e-5, 0.1]  | uniform |
|         | momentum      | [0.1, 0.99]  | uniform |
|         | max_dropout   | [0.0, 1.0]   | uniform |

`logfinrange` in Syne Tune, which are encoded by a single integer internally. For LCBench, we run the 5 most expensive tasks among the 35 tasks available ("airlines", "albert", "covertype", "christine" and "Fashion-MNIST").

In Section 4.1, all results are obtained by running simulation on a c5.4xlarge machine. All schedulers are run with 4 workers a wallclock time budget of 1200s for FCNet, 21600s for NAS201 and 7200s for LCBench. For LCBench, we also stop the tuning whenever 4000 epochs evaluations are seen given that the epoch-time varies more across tasks. As indicated in Section 3.2, those results include the decision time taken by schedulers to suggest new configuration or decide on stopping (longer for GP than ASHA for instance) as well as the time taken by each configuration to be evaluated (as recorded in the tabulated benchmarks). Since LCBench does not contain all possible evaluations on a grid, we run evaluations using a $k$-nearest-neighbors surrogate with $k = 1$.

Confidence intervals are reported using mean and standard deviation with 30 random seeds in Figures 2 and 4. To compute normalized ranks in Table 2, we first compute the best value obtained for each methods at 10 uniformly spread time-steps from 0 to the maximum allowed wallclock time set for the task. We then compute the normalized ranks for each method in [0, 1] and average those values over time-steps and tasks of the benchmark.

**Schedulers details**. We give here the list of parameters used for running schedulers in our experiments. In general, the Syne Tune defaults have been used.

- REA is run with a population size of 10, and 5 samples are drawn to select a mutation from

- GP is run using a Matérn $\frac{5}{2}$ kernel with automatic relevance determination parameters. For each suggestion, the surrogate model is fit by marginal likelihood maximization, and a configuration

is returned which maximizes the expected improvement acquisition function. This involves averaging over 20 samples of fantasy outcomes for pending evaluations (Snoek et al., 2012).

- TPE is based on a multi-variate kernel density estimator as proposed by Falkner et al. (2018) to capture interactions between hyperparameters, which is not possible with unit-variant kernel density estimator as used for the original TPE approach (Bergstra et al., 2011). We limit the minimum bandwidth for the kernel density estimator to 0.1 to avoid that all probability mass is assigned to a single categorical value, which would eliminate extrapolation.

- MSR is running random-search with the median stopping rule, trials are stopped at a time-step if their running average is worse than the median observed for this time-step. The stopping only occurs after a grace time of 1 and when at least 5 results are observed for a given time-step.

- ASHA is running the stopping variant (see Section 3.3) with grace period 1 and reduction factor 3, so that stopping trials happens after $1, 3, 9, \ldots$ epochs. Configurations for new trials are sampled at random.

- BOHB uses the same multi-variate kernel density estimator as TPE, and hyperparameters are set to default values in Falkner et al. (2018). Note that BOHB uses the same asynchronous scheduling as ASHA and MOB, while the algorithm in Falkner et al. (2018) is synchronous (corresponding to SyncBOHB in Syne Tune).

- MOB is running the same scheduling as ASHA, but configurations for new trials are chosen as in Bayesian optimization. The surrogate model is adapted from (Swersky et al., 2014), but does not use their conditional independence assumption (the base kernel over $x$ is Matérn $\frac{5}{2}$ ARD). We deal with pending evaluations by averaging the acquisition function over 20 samples of fantasy outcomes (Snoek et al., 2012). Details are found in (Klein et al., 2020).

- ASHA-BB runs ASHA on a restricted configuration space that consists in the bounding-box of the best evaluations of offline tasks. The same parameters are used as for ASHA and we use one-hot encoding to compute the bounding-box of categorical parameters.

- ASHA-CTS uses the same parameters as ASHA but use copula-thompson-sampling to find configurations to evaluate as proposed in (Salinas et al., 2021b). This approach requires to fit a surrogate on offline evaluations which is done with XGBoost with default hyper-parameters.

- RUSH uses the same settings as ASHA but uses the best configuration from each other task as first candidates and all candidates which don't exceed their performance at the different rung levels are immediately pruned.

- ZS is the method Wistuba et al. (2015b) refer to as A-SMFO. This method is free from any parameters. This approach requires to fit a surrogate on offline evaluations for benchmarks where configurations are not evaluated for all datasets (FCNet) which is done with XGBoost with default hyper-parameters.

## C  Additional results

In Figure 4, we show the performance obtained on all methods and tasks.

## D  Code examples

### D.1  Reporting metrics and tuning script conventions

In Listing 2, we show an example of how a user can report metrics from a training script. In l15, the user reports a metric of a given validation loss for an epoch. Additional metric such as runtime, dollar-cost (when running on cloud machines) are also automatically added.

To support checkpointing the user can write into the folder `st_checkpoint_dir` passed as argument in l7. The trial backend makes sure that those folders are unique and properly persisted so that they can be accessed between trials for advanced schedulers such as PBT.

## D.2 Switching between different trial backends

**Amazon SageMaker**. Listing 1 showed how to run a tuning on a local machine. How to run on GPU cloud machines with distributed training is shown in Listing 3. Importantly, this can be achieved by just passing a different `trial_backend` to the tuner. Users can also include additional dependencies by providing a "requirements.txt" file, provide custom code or their docker image.

**Simulation backend**. To evaluate schedulers, we provide a simulation backend that allows to run realistic simulations in a fraction of the wallclock time and cost compared to real experiments or compare to a naive approach consisting in sleeping for the recorded runtime of each configuration. An example on how to use this backend is shown in Listing 4. As for switching to cloud backend, the only change required is to pass a different trial backend to the tuner. A surrogate can be used by just passing arguments to the class, for instance using a KNN surrogate can be achieved by setting `trial_backend = BlackboxRepositoryBackend(blackbox_name="lcbench",` `dataset="Fashion-MNIST", surrogate="KNeighborsRegressor")` which is the setup that was used for LCBench.

## D.3 Launching tuning remotely

The Listing 5 shows how a user can easily launch a tuning remotely using any desired machine. The main utility is the ability to schedule many experiments in parallel (for instance when benchmarking one algorithm multiple seed/task combinations must be evaluated). Importantly, a user can also use the Amazon SageMaker backend in this scenario (in which case the remote machine schedules Amazon SageMaker training jobs for different trials). The cost is detected either when the user tunes remotely or when he uses the Amazon SageMaker backend. In addition to letting the user the possibility to cap the dollar budget of a tuning, it can also be used as an objective metric to minimize.

```python
# train_network.py
from syne_tune import Reporter
from argparse import ArgumentParser

if __name__ == '__main__':
    parser = ArgumentParser()
    parser.add_argument('--st_checkpoint_dir', type=str)
    parser.add_argument('--epochs', type=int)
    parser.add_argument('--num_layers', type=int)
    parser.add_argument('--learning_rate', type=float)
    args, _ = parser.parse_known_args()

    report = Reporter()
    for i in range(args.epochs):
        val_loss = train_epoch()
        report(val_loss=val_loss, epoch=i+1)  # Feed the score back to Syne Tune.
```

Listing 2: Changing the trial backend to run on cloud machines

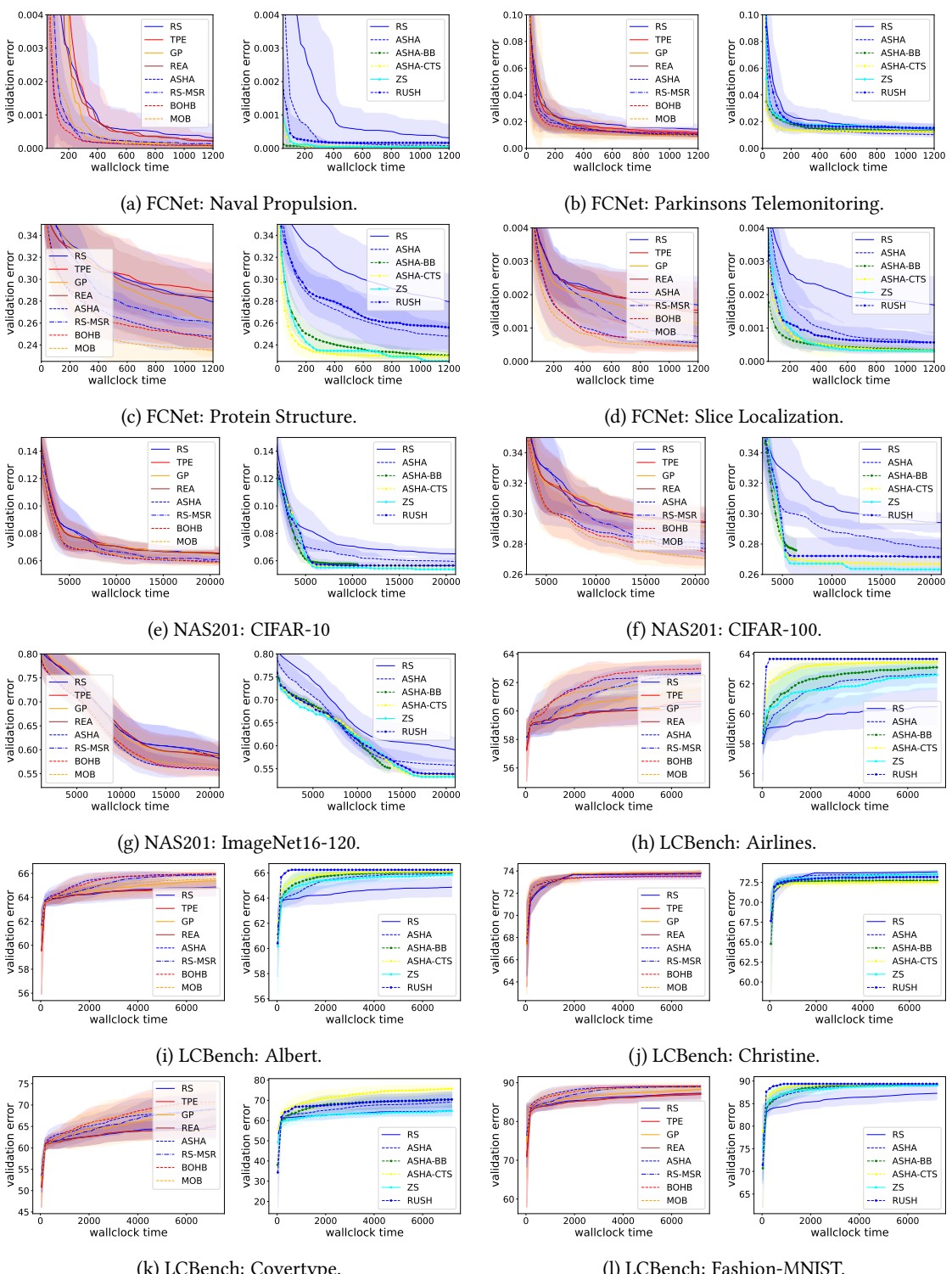

(a) FCNet: Naval Propulsion.

(b) FCNet: Parkinsons Telemonitoring.

(c) FCNet: Protein Structure.

(d) FCNet: Slice Localization.

(e) NAS201: CIFAR-10

(f) NAS201: CIFAR-100.

(g) NAS201: ImageNet16-120.

(h) LCBench: Airlines.

(i) LCBench: Albert.

(j) LCBench: Christine.

(k) LCBench: Covertype.

(l) LCBench: Fashion-MNIST.

Figure 4: Best validation error found over time on all tasks.

```python
1   from sagemaker.pytorch import PyTorch
2   from syne_tune import Tuner, StoppingCriterion
3   from syne_tune.backend import SageMakerBackend
4   from syne_tune.config_space import randint, loguniform
5   from syne_tune.backend.sagemaker_backend.sagemaker_utils import get_execution_role
6   from syne_tune.optimizer.baselines import BayesianOptimization
7
8   # hyperparameter search space to consider
9   config_space = {
10      'epochs': 100,
11      'num_layers': randint(1, 20),
12      'learning_rate': loguniform(1e-6, 1e-4)
13  }
14
15  # use SageMaker trial backend to evaluate trials on the cloud
16  # using pre-build PyTorch container
17  trial_backend = SageMakerBackend(
18      sm_estimator=PyTorch(
19          entry_point='train_network.py',
20          instance_type='ml.p2.xlarge',  # choose a GPU machine
21          instance_count=10,  # run distributed training with 10 nodes
22          role=get_execution_role(),
23          framework_version='1.7.1',
24          use_spot_instances=True,
25          py_version='py3',
26      )
27  )
28
29  tuner = Tuner(
30      trial_backend=trial_backend,
31      scheduler=BayesianOptimization(config_space, metric='val_loss'),
32      stop_criterion=StoppingCriterion(max_wallclock_time=30),
33      n_workers=4,  # how many trials are evaluated in parallel
34  )
35  tuner.run()
```

Listing 3: Changing the trial backend to run on cloud machines

```
1   from syne_tune.config_space import randint, loguniform
2   from benchmarking.blackbox_repository.simulated_tabular_backend import BlackboxRepositoryBackend
3   from syne_tune import Tuner, StoppingCriterion
4   from syne_tune.optimizer.baselines import BayesianOptimization
5
6   # hyperparameter search space to consider
7   config_space = {
8       'epochs': 100,
9       'num_layers': randint(1, 20),
10      'learning_rate': loguniform(1e-6, 1e-4)
11  }
12
13  trial_backend = BlackboxRepositoryBackend(
14      blackbox_name="nasbench201",
15      dataset="cifar100",
16  )
17
18  # runs parallel tuning with simulations
19  tuner = Tuner(
20      trial_backend=trial_backend,
21      scheduler=BayesianOptimization(config_space, metric='val_loss'),
22      stop_criterion=StoppingCriterion(max_wallclock_time=30),
23      n_workers=4,   # how many trials are evaluated in parallel
24  )
25  tuner.run()
```

Listing 4: Changing the trial backend to run on parallel and asynchronous simulations based on tabulated benchmarks.

```
1   from syne_tune import Tuner, StoppingCriterion
2   from syne_tune.config_space import randing, loguniform
3   from syne_tune.backend import LocalBackend
4   from syne_tune.optimizer.baselines import BayesianOptimization
5   from syne_tune.remote.remote_launcher import RemoteLauncher
6
7   # hyperparameter search space to consider
8   config_space = {
9       'epochs': 100,
10      'num_layers': randint(1, 20),
11      'learning_rate': loguniform(1e-6, 1e-4)
12  }
13
14  tuner = RemoteLauncher(
15      tuner=Tuner(
16          trial_backend=LocalBackend(entry_point='train_network.py'),
17          # do not spend more than 10$ or 600s
18          stop_criterion=StoppingCriterion(max_cost=10, max_wallclock_time=600),
19          scheduler=BayesianOptimization(config_space, metric='val_loss'),
20      ),
21      instance_type='ml.p3.8xlarge',   # runs on a remote machine with 8 GPUs
22      use_spot_instances=True,
23  )
24  tuner.run()
```

Listing 5: Running a tuning on a remote instance

