# OpenReview forum: "Syne Tune: A Library for Large Scale Hyperparameter Tuning and Reproducible Research"
_automl.cc/AutoML/2022/Track/Main — AutoML-Conf 2022 (Main Track)_

### Official Review · Reviewer_SBjV · 2022-03-27

**Potential Impact On The Field Of Automl Rating:** 3
**Technical Quality And Correctness:** The technical quality is good and det…
**Technical Quality And Correctness Rating:** 4
**Clarity:** The presentation is clear with high l…
**Clarity Rating:** 3

**Summary Of Contributions:**

This work presents Syne Tune, a backend agnostic HPO system, that contains implemented baselines and supports hyperparameter transfer learning, constrained HPO and multi-objective HPO. The paper presents the design of system and provides a benchmark for different HPO algorithms (asyn, syn), which is valuable to the community.

**Overall Review:**

See the summary below.

**Potential Impact On The Field Of Automl:**

Due to the rich algorithms supported and benchmark presented in the paper, the community would surely benefit from the work. This system also allows the HPO users to compare and implement new HPO algorithms.

**Reproducibility:**

This package is open sourced on Github, with documentations.

**Review Confidence:**

4: You are confident in your assessment, but not absolutely certain. It is unlikely, but not impossible, that you did not understand some parts of the submission or that you are unfamiliar with some pieces of related work.

**Review Rating:**

5: Accept, good paper

**Review Summary:**

This work introduces Syne Tune, another package in HPO. The presentation is clear and sound. But several issues should be better clarified.

Advantage:
- Syne Tune contains implemented baselines and supports hyperparameter transfer learning, constrained HPO and multi-objective HPO.
- The paper introduces a benchmark of different algorithms.
- Good illustration cases are introduced.

Disadvantage:
- The design part is too vague. It is not clear how in architecture design, API, Syne Tune differs from other works.
- The related work section already mentions a lot of HPO packages. The authors should elaborate more why we need another packge on HPO. Does it support something which is not at all supported by other packages? For example, from what I understand, Optuna covers the functionality of Syne Tune, the only difference is that Optuna doesn't come up with built-in most recent methods (but should totally implementable) Thus,  I'd suggest a revised part on related works to discuss more thoroughly (since we already have a lot packages...)
- Some features are a bit over-claimed, e.g. multi-objective (which in Syne-Tune, is mostly choose machine types on AWS)

---

### Official Review · Reviewer_JVbt · 2022-04-05

**Potential Impact On The Field Of Automl Rating:** 4
**Technical Quality And Correctness Rating:** 4
**Clarity:** The paper is well written and clear.
**Clarity Rating:** 4

**Summary Of Contributions:**

This paper present a library for hyperparameter optimization (HPO) called Syne Tune. This library offers a wide set of HPO algorithms, from more traditional algorithms, to multi-fidelity, optimization with constraints and even transfer learning. The library also provides extensive benchmarking tasks with pre-built tabular datasets of the tasks to speedup benchmarking. The benchmarking system also takes into account the relative timing of trainings to simulate long training tasks and benchmark parallelization of HPO algorithms. The author provide 3 use cases to show-case the utility of the library, one benchmarking asynchronous HPO algorithms, one assessing the effectiveness of a algorithm for optimization with constraints (cBO), and one providing an example of multi-optimization with MOASHA.

**Overall Review:**

+ The library has nice variety of algorithms
+ The library supports algorithms for transfer learning in HPO, which I haven't seen yet in any other library
+ The benchmarking system seems very helpful, especially how it can help simulate parallelized trainings on long training. Tabular datasets is helpful, but this could easily be reused outside the library.

- Since many of the algorithms seems to be in-house reimplementations, it would have been great to have performance comparison to previous implementations (Hyperband/ASHA/PBT from Ray-tune, TPE from Hyperopt and Optuna, BO compared to DragonFly and BoTorch/Ax). This would be a great contribution for a paper.

**Potential Impact On The Field Of Automl:**

This paper could have a profound impact on the field if it is widely adopted. As pointed out by the authors, there as not been wide adoption of HPO libraries yet by the community of ML researchers. Some libraries are more populare but nowhere as popular as PyTorch or TensorFlow. This library seems to offer convenient modularity of algorithm definition as well as a very useful benchmarking system. It's support for multiple backends for the parallelization could also be very helpful for wide adoption. I am concerned however that it remains some form of vendor lock-in as long as the main multi-node parallelization is only supported through AWS.

**Reproducibility:**

The authors did not include a reproducibility check list, and as is, the experiments are not reproducible. I do not believe it is critical for this paper as the experiments only serve as use case examples. I would trust the authors to provide similar examples in their library documentation however to help users getting started.

**Review Confidence:**

5: You are absolutely certain about your assessment. You are very familiar with the related work and checked all the details carefully.

**Review Rating:**

5: Accept, good paper

**Review Summary:**

The library presented in this paper could be very useful for researchers developing new HPO algorithms. The apparent modularity of the algorithms implementation and the benchmarking system seams to be especially useful. The use case examples are clear and concise. I believe this paper is a very appealing presentation for this library.

**Technical Quality And Correctness:**

This paper covers the main feature of the library and therefore does no go deep into technical details. The experiments are only use-case demonstration and does not aim support any strong claim.

I would be curious to hear from the authors why they favored tabular datasets over simulation based on surrogate models.

---

### Official Review · Reviewer_2Gkn · 2022-04-05

**Potential Impact On The Field Of Automl Rating:** 3
**Technical Quality And Correctness Rating:** 3
**Clarity:** I found the paper clear and easy to u…
**Clarity Rating:** 4
**Ethics Rating:** Yes, research integrity issues (e.g.,…

**Summary Of Contributions:**

The paper describes a hyper-parameter optimization library called SyneTune.
Different from existing libraries, SyneTune provides multiple methods for HPO like random search, Bayesian opt, and evolutionary search. It additionally supports HPO for transfer learning, constrained optimization and multi-objective optimization. There is also support for benchmarking, similar to HPOBench.
The authors provide minimal code examples for calling their API for HPO (in the main paper and supplementary). The experiments demonstrate HPO results showcasing the above capabilities: comparing multiple algorithms and transfer learning, constrained HPO, and multi-objective HPO.
Working code samples or a python notebook would have made the paper more impactful.

**Ethics Details (Optional):**

The code link in the supplementary is non-anonymous and reveals authors identity.
Most of the review was written without this information and the reviewer tried to be objective post seeing this information.

————-
The authors comment addressed this issue.

**Overall Review:**

Pros:
- Interesting library providing many interesting features for HPO in one place.

Cons:
- There are many existing libraries for HPO already and incentives to switch to a new one can be low.

**Potential Impact On The Field Of Automl:**

I think the package could be useful for reproducible AutoML research in the future.

**Reproducibility:**

It seems easy enough to reproduce the results, though a demo through a colab or jupyter notebook would have been more useful.

**Review Confidence:**

2: You are willing to defend your assessment, but it is quite likely that you did not understand the central parts of the submission or that you are unfamiliar with some pieces of related work.

**Review Rating:**

5: Accept, good paper

**Review Summary:**

Overall I found the presentation clear, the analysis sufficient and the proposed library interesting to warrant consideration to use it in future HPO work.

**Technical Quality And Correctness:**

The paper describes a new library and showcases results of running it on several benchmarks under different settings. I did not find any technical issues.

---

### Official Review · Reviewer_BqTn · 2022-04-05

**Potential Impact On The Field Of Automl Rating:** 4
**Technical Quality And Correctness Rating:** 3
**Clarity Rating:** 4

**Summary Of Contributions:**

The paper introduces a library that provides various components necessary for performing hyperparameter optimization, as well as doing benchmarks on HPO methods. The library comes with a variety of HPO-methods that offer multi-fidelity optimization, constrained optimization, multi-objective optimization, and transfer learning capabilities. The library furthermore provides access to established tabular HPO benchmarks. A demonstration run of the included methods on some included benchmarks is presented that show the benefits of multi-fidelity and transfer learning methods.

**Clarity:**

The work is presented very clearly. Very minor things I may not have understood well:
* l. 131-132: what is the 'least busy GPU'? Measured by free GPU RAM?
* l. 156 Does this sentence mean that there is no single-worker sequential GP scheduler included that proposes a new configuration for evaluation only after the single worker is done?

**Overall Review:**

The work is of high quality and explains the the presented library well. The library appears relevant and useful.

A few further comments / questions:
* Has the technique presented in lines 143-149, to the knowledge of the authors, already been used by other work? In that case it should be cited. Otherwise it may be worth stressing that this is an original contribution of the work.
* Typo in caption of Table 1: "a random configurations"


**Potential Impact On The Field Of Automl:**

The presented library includes many HPO methods that I would consider state of the art and can therefore be useful for practical HPO tasks; in particular since it stresses the practically relevant asynchronous many-worker (possibly cloud) setting. The inclusion of different benchmarks is also a benefit for other researchers that could attach their own HPO methods to the library: they would have an easy way of benchmarking their new methods against established HPO methods.

**Reproducibility:**

The reproducibility list is included and filled out. Formal issue: the code link appears to break blinding.

**Review Confidence:**

3: You are fairly confident in your assessment. It is possible that you did not understand some parts of the submission or that you are unfamiliar with some pieces of related work.

**Review Rating:**

6: Strong accept, should be highlighted

**Review Summary:**

The paper is of high quality and presents a library that appears relevant and useful. There are very few clarifications that I would suggest the authors to make.

**Technical Quality And Correctness:**

The work is of high technical quality. The only minor point I would raise is that the four orders of magnitude runtime difference between the Syne Tune's and HPOBench's implementation of FCNet seems implausible (Table 1). I am not an HPOBench user, but I can imagine there are ways to use it that are more or less efficient; was HPOBench used in a different way than it should ordinarily be used? E.g. did the comparison not make use of possible batch-evaluations of the objective?

---

### Official Review · Reviewer_WeeR · 2022-04-07

**Potential Impact On The Field Of Automl Rating:** 4
**Technical Quality And Correctness Rating:** 4
**Clarity Rating:** 4

**Summary Of Contributions:**

This paper introduces a new library for executing and benchmarking hyperparameter optimization algorithms. The library allows for different back-ends: local, cloud, and simulation, and facilitates distributed, asynchronous optimization. The library implements several popular algorithms for suggesting configurations: e.g., random search, Bayesian optimization, TPE, as well as various schedulers for multi-fidelity, transfer, multi-objective, and constrained scenarios. The library also includes a number of benchmarks: FCNet, NAS201, and LCBench.

**Clarity:**

The paper was easy to read and understand. It is well organized, clearly labelling the different facets of the library, their purpose, and highlighting the different example that come with the library for each. I think that the level of detail for the different baselines and schedules is appropriate for a paper like this.

**Overall Review:**

This is a nicely written library that implements many common and contemporary baselines, schedulers, and benchmarks for hyperparameter optimization. Based on the experiments, it appears to be significantly faster than alternative implementations, while also being far more broad and adaptable. I think that this could easily see usage either to facilitate research in the area, or as a tool for launching HPO experiments with existing approaches.

In terms of the baselines (the part that suggests a new configuration) I am a bit torn. This library implements a push model, meaning it is responsible for launching experiments and scheduling them. An alternative is a pull model, where a server is queried for configurations, and the launch details are left to the user. I prefer the latter for getting new configurations, but I appreciate that the push model makes more sense for the implemented schedulers. As long as the local scheduler and cloud schedulers are robust and continue to be maintained, then I don't see this as being an issue.

**Potential Impact On The Field Of Automl:**

The API is clean, and the number of baselines, schedulers, back-ends, and benchmarks is exhaustive. I think that this could be very useful in facilitating the development of new HPO techniques. The main question is whether researchers will find it easier to use/extend the library, or implement their own setup.

At the very least, with the provided options it can be very useful for practitioners, especially with its focus on the distributed/cloud setting.

**Reproducibility:**

A link to a Github repo with the code is provided, along with specific code examples of using the API. The examples folder in the repo is extensive and does appear to reproduce most of the plots in the paper, although I have not run the code myself to verify.

**Review Confidence:**

4: You are confident in your assessment, but not absolutely certain. It is unlikely, but not impossible, that you did not understand some parts of the submission or that you are unfamiliar with some pieces of related work.

**Review Rating:**

5: Accept, good paper

**Review Summary:**

See the Overall Review section.

**Technical Quality And Correctness:**

Technically, the main contributions are the API and the experiments. The API examples are clear, and the code looks clean.

The experiments are mainly to a) compare to other similar libraries and show how Syne Tune is more efficient, b) showcase some of the implemented algorithms and schedulers for a brief comparison and to highlight the versatility of the library. This also provides a sanity check to ensure that the baselines reproduce the expected benchmark results.

---

### Meta-Review · Area_Chair_fUru · 2022-05-09

**Recommendation:** Accept
**Confidence:** 4

**Metareview:**

Reviewers were satisfied with the empirical evaluation, description of architecture, and reproducibility. The unanimously agreed that this work should be accepted.

---

### Decision · Program_Chairs · 2022-05-13

Accept